# Cybersecuity Analysis of a Telemedicine Platform

**DOI:** 10.3390/healthcare13020184

**Published:** 2025-01-18

**Authors:** Martina Nobili, Domenico Raguseo, Roberto Setola

**Affiliations:** 1Unit of Automatic Control, Universitá Campus Bio-Medico di Roma, 00128 Rome, Italy; 2Exprivia, 70056 Molfetta, Italy; domenico.raguseo@exprivia.com

**Keywords:** risk assessment, AHP, cybersecurity, telemedicine framework, medical data protection

## Abstract

**Background:** The global shift toward telemedicine, accelerated by the COVID-19 pandemic, has revolutionized healthcare delivery by enabling remote consultations and treatments. However, this rapid adoption has also introduced critical cybersecurity vulnerabilities, particularly in safeguarding sensitive medical data and ensuring the secure operation of telemedicine platforms. If not properly addressed, these vulnerabilities can compromise patient safety and the integrity of healthcare systems. As a result, implementing robust cybersecurity measures in telemedicine platforms is essential. **Methods:** The framework developed in this study allows for the assessment of a telemedicine platform’s cybersecurity posture and provides concrete recommendations for improvement. In this context, the Security Framework for Telemedicine Platforms, developed as part of the study, serves as a valuable tool for evaluating platform security, identifying vulnerabilities, and pinpointing areas for enhancement. **Conclusions:** This framework empowers organizations to effectively strengthen their cybersecurity strategies, as demonstrated by a case study.

## 1. Introduction

The spread of technology within the healthcare sector is one of the most significant phenomena of recent decades, leading to major changes not only in medical practice, with the introduction of new diagnostic technologies, but also in healthcare management and patient experience. The COVID-19 pandemic accelerated the digital transition, particularly in healthcare. The impossibility of direct patient contact forced the nation’s health services to rapidly innovate and implement remote care solutions, such as Electronic Health Records (EHRs), connected medical devices, and telemedicine platforms. According to data from the Milan Polytechnic’s Digital Innovation in Health Care Observatory, in 2019 only 4% of specialist physicians and 3% of general practitioners were using telemedicine solutions, while in 2020, three out of four specialist physicians said the tool played an instrumental role during the emergency phase [1,2]. Furthermore, according to [3], telemedicine consultations in Italy increased by more than 200% in the first months of the pandemic.

In Italy, the National Plan for Recovery and Resilience (PNRR) highlights the growing diffusion and innovative potential of telemedicine. Under the Health Mission of the PNRR (M6C1), there is an objective to promote and integrate telemedicine services and performances as a structural component of the national health system (NHS) to support patients with chronic diseases. Initially, EUR 1 billion was allocated to fund projects enabling remote doctor-patient interactions and ad hoc research initiatives on digital technologies in telemedicine [4]. Following the reshaping of the NRP, resources for telemedicine increased by EUR 500 million, raising the final target (Target M6C1-9) by 100,000 people for a total of 300,000 assisted with telemedicine services by the 2025 deadline [2].

According to the definition provided by the Ministry of Health, Italy—“Telemedicine National Guidelines” (2012): “Telemedicine refers to a mode of delivery of healthcare services, through the use of innovative technologies, particularly Information and Communication Technologies (ICT), in situations where the health professional and the patient (or two professionals) are not in the same location. Telemedicine involves the secure transmission of medical information and data in the form of text, sound, images, or other forms necessary for the prevention, diagnosis, treatment, and subsequent follow-up of patients. Telemedicine services should be equated with any diagnostic/therapeutic health service”.

The Ministry of Health has issued guidelines to standardize telemedicine practices across the country, ensuring consistency in service delivery and patient care [1]. These regulations have been crucial in legitimizing telemedicine and encouraging its widespread use.

To support the activities and services offered through telemedicine, telemedicine platforms have emerged. These online systems or mobile applications integrate various features for remote health diagnosis and monitoring, providing tools such as video calls, instant messaging, and e-mail to facilitate doctor-patient communication. They also include health information management systems that enable healthcare providers to access patients’ clinical information, track their treatments, and schedule follow-up visits effectively [5]. This highlights the role of advanced telemedicine platforms in improving the quality and efficiency of remote consultations. Telemedicine platforms are considered software medical devices, as they meet the definition of a medical device provided by Article 2(1) of the EU Medical Device Regulation (MDR) No. 2017/745.

Despite the benefits for patients, the widespread use of technology in healthcare introduces new challenges, particularly regarding cybersecurity. The sector faces increasing risks of data breaches and cyberattacks targeting medical software and Internet of Things (IoT) devices, which can compromise both system security and patient safety. According to Clusit’s (Italian Association for Information Security) 2024 Report on ICT Security in Italy, globally, the healthcare sector is the sector most affected by cyberattacks, second only to the “Multiple Targets” category. Furthermore, cyberattacks targeting the healthcare sector are showing a rapidly growing trend, with numbers doubling from 2022 to 2023 [6]. An analysis conducted by ENISA (European Union Agency for Cybersecurity) covering January 2021 to March 2023 shows that in Europe, hospitals were the facilities most affected by cyberattacks, suffering 42% of total incidents [7]. Additionally, France, Spain, and Italy emerged as the nations with the highest number of cyberattacks. However, this finding is difficult to interpret as it may depend on various factors, such as population size and differences in cyber incident reporting capacity or data collection processes among countries [7]. The same report reveals that the main target of attacks in the healthcare sector is often sensitive personal data. Cybercriminals commonly use malware attacks to threaten the dissemination of information and extort money [7].

Ref. [8] discusses common vulnerabilities, including weak authentication protocols, inadequate encryption, and insufficient security updates. To address these security issues, the Italian government has strengthened regulatory frameworks for telemedicine. The General Data Protection Regulation (GDPR) and other local regulations mandate stringent data protection measures [9]. This highlights the importance of compliance with these regulations to safeguard patient privacy and data security.

Healthcare providers have been urged to implement comprehensive cybersecurity strategies, including the adoption of advanced encryption methods, multi-factor authentication, and regular security audits [10]. This emphasizes the need for continuous monitoring and updating of security protocols to counter evolving cyber threats. Raising awareness and training healthcare professionals on cybersecurity best practices is critical, as many securities breaches stem from human error, such as phishing attacks and weak passwords [11]. There is a need to advocate for regular training programs to educate staff on recognizing and responding to cyber threats. Healthcare institutions are increasingly collaborating with cybersecurity experts to enhance their defenses. These experts provide valuable insights into the latest threat landscapes and recommend tailored security solutions [12]. As such, there is a need to discuss successful collaborations between hospitals and cybersecurity companies in mitigating risks.

This paper will present a security framework for telemedicine platforms. To present and implement this work, the specific requirements that these platforms must meet are analyzed to integrate them into a single model. The paper is structured as follows: in Section 2, the state of the art is reviewed; in Section 4, our framework is introduced; next, a case study is presented in Section 5; finally, conclusions are given in Section 7.

## 2. State of the Art

Several cybersecurity frameworks were developed to answer the risks to which medical devices are subjected as a result of their connection to the network. In [13], a model is proposed that integrates various models operating across different security domains. This paper examines the advancements in Medical Cyber-Physical Systems (MCPS) from 1998 to 2020 and introduces a novel security and privacy framework that incorporates multiple security models. The proposed framework, evaluated qualitatively, offers significant benefits to healthcare sectors utilizing smart devices, enhancing security in Internet of Things (IoT) environments, optimizing patient workflow operations, and establishing robust cybersecurity infrastructures within healthcare applications.

In [14], the authors introduced a comprehensive blockchain-enabled cybersecurity architecture for healthcare informatics, ensuring the privacy, reliability, and legal compliance of patient data. Utilizing AES-Diffie-Hellman for secure communication, blockchain with Proof-of-Work (PoW) for immutable data storage, and Role-Based Access Control (RBAC) for granular access management, the framework addresses key security challenges. The architecture ensures secure data transmission and storage, significantly reducing unauthorized access incidents by 97.9%. Implemented in Python, the framework outperforms NTRU, RSA, and DES with encryption and decryption times of 12.1 and 12.2 s, respectively.

Ref. [15] proposes a hybrid cybersecurity framework, extending NIST version 1.1, adapted to the adoption of IoMT by the UK healthcare sector, considering current cybersecurity frameworks such as ISO 27000 [16], NIST CSF 2018 [17], and COBIT [18] to be insufficient for the specific needs of IoMT.

Unlike the previously presented frameworks, our proposed model will provide a quantitative view to make an assessment of cyber security measures. In particular, we will focus on the analysis of security measures proposed within telemedicine platforms.

### Medical Regulations

Given the sensitivity of the data and instrumentation, several regulations must be followed. The key safety-related regulations particularly related to medical devices are also presented.

According to the Ministry of Health, the market introduction, availability, and operation of medical devices in Italy are allowed for devices bearing the CE mark, demonstrating compliance with the applicable requirements of Regulation (EU) 2017/745 [19], known as the Medical Devices Regulation (MDR 745:2017). This regulation aims to ensure a robust, sustainable regulatory framework with transparent procedures to maintain high safety levels [20]. The MDR is crucial for the development of telemedicine platforms as it establishes essential safety requirements, including minimum cybersecurity requirements, for all medical devices incorporating programmable electronic systems. Key aspects of the MDR that can enhance the cybersecurity of telemedicine platforms include Risk-Based Approach, Device Manufacturing and Environmental Interaction, Lifecycle Management of Software, Incident Reporting, Training, and Awareness.

To fulfill the requirement introduced by Medical Device Regulation, many manufacturers adopt the EN ISO 13485 standard [21], which is the specific Quality Management System (QMS) standard tailored for medical device companies. This standard provides a practical foundation for addressing the MDR’s requirements. Additionally, the standard necessitates the evaluation and monitoring of suppliers for components and services used in the production of medical devices. This requirement can be extended to include the assessment of software and IT service providers, ensuring they adhere to relevant cybersecurity standards and thereby reducing the risk of vulnerabilities within the device ecosystem. A core principle of ISO 13485 is the continuous identification and correction of issues, which includes cybersecurity-related problems. Continuous improvement processes ensure that any detected vulnerabilities are promptly addressed and that preventive measures are implemented. Moreover, ISO 13485 emphasizes the importance of training personnel involved in the QMS. Incorporating cybersecurity awareness and best practices into this training can significantly reduce the risks associated with human error. Although ISO 13485 does not provide specific guidelines for managing cybersecurity incidents, it does require the implementation of procedures for identifying problems and taking corrective actions. This framework can be adapted to address cybersecurity anomalies and ensure that preventive actions are taken to avoid future incidents.

ISO 14971, “Medical Devices–Application of Risk Management to Medical Devices”, [22] is an international standard that specifies terminology, principles, and processes for risk management related to medical devices, including software used as a medical device. The standard provides crucial definitions such as “benefit”, “hazard”, “harm”, “risk”, “residual risk”, “risk acceptability”, and “safety”, establishing a foundational language for risk management. Moreover, general requirements for the risk management system are detailed. Manufacturers are required to establish, implement, document, and maintain a continuous process for identifying hazards and hazardous situations associated with a medical device, estimating and evaluating the associated risks, controlling these risks, and monitoring the effectiveness of the risk control measures. This process applies throughout the entire lifecycle of the medical device. The risk management process must include risk analysis, risk evaluation, risk control, and production and post-production activities. Furthermore, the 2019 edition of ISO 14971 introduced significant updates, including the integration of cybersecurity into risk management for medical devices. Manufacturers are now required to identify cybersecurity-related hazards, assess the associated risks, and implement appropriate control measures to mitigate these risks. This integration ensures that cybersecurity threats are systematically addressed, enhancing the overall safety and security of medical devices.

IEC 62304 [23], “Medical Device Software–Software Life Cycle Processes”, sets international standards for the development and lifecycle management of medical device software. It categorizes software into safety classes based on potential impact (Class A, B, C), determining development rigor. Processes include planning, requirements analysis, design, implementation, testing, and maintenance, emphasizing documentation and verification. However, it lacks explicit cybersecurity directives, critical in healthcare’s vulnerability to cyber threats.

IEC 81001-5-1 [24] addresses this gap, focusing on integrating cybersecurity across software design, development, and lifecycle management. Manufacturers must define security requirements, conduct rigorous testing, and implement controls like encryption and access management. Continuous updates and user training are essential for maintaining secure telemedicine platforms, ensuring patient safety and system reliability.

In summary, these standards provide a guideline for developing and maintaining secure medical devices with the final aim being to ensure patient safety and system reliability. Unfortunately, none of them are specifically tailored for telemedicine platforms. Moreover, the standard only partially overlaps each other because each of them emphasize different peculiar aspects of cybersecurity issues.

The General Data Protection Regulation (GDPR) [25], EU Regulation 679/16, effective from May 2018, mandates strict rules for processing personal data to protect individuals’ rights and freedoms. It requires data to be stored only as long as necessary for their purpose, promoting data minimization and security through technical and organizational measures. GDPR necessitates data controllers to maintain records of processing activities (Article 24) and integrate “Privacy by Design” and “Privacy by Default” into product and service development (Article 25). Data processors must implement appropriate safeguards (Article 28), and both controllers and processors must adopt measures like pseudonymization and encryption (Article 32) to ensure data security, confidentiality, integrity, availability, and resilience. The regulation also mandates prompt notification of personal data breaches (Articles 33 and 34) to supervisory authorities and affected individuals, ensuring transparency and mitigation of breach impacts. Regular testing and evaluation of security measures are required to maintain data processing security.

Integrating these regulatory requirements with ISO 27100’s [26] guidelines for an Information Security Management System (ISMS) and the principles of the National Framework for Cybersecurity and Data Protection ensures a robust security and compliance strategy. ISO/IEC 27100 specifies requirements for establishing, implementing, maintaining, and continually improving an ISMS, emphasizing confidentiality, integrity, and availability of information. This standard outlines organizational context assessment, leadership involvement, risk management, performance evaluation, and continuous improvement, supported by Annex A’s detailed security controls.

The Italian national Framework for Cybersecurity and Data Protection [27] further supplements this by structuring cybersecurity measures into identify, protect, detect, respond, and recover functions, each with specific categories and subcategories that guide comprehensive risk management and incident response. This integration of GDPR, ISO 27100, and the Italian national Framework for Cybersecurity and Data Protection provides a comprehensive approach to managing data security and regulatory compliance, essential for safeguarding sensitive health data and maintaining the integrity of telemedicine platforms.

The ioXt Alliance (Internet of Secure Things Alliance) [28] is a global industrial consortium dedicated to developing and promoting guidelines, standards, and certification programs aimed at enhancing IoT device security to protect consumer data privacy, security, and integrity. The ioXt Alliance includes device manufacturers, technology providers, network operators, regulatory bodies, and other organizations committed to advancing IoT security and reliability. The ioXt Alliance’s security requirements are based on the eight principles of the ioXt Security Pledge. Each pledge can have multiple requirements and varying security levels depending on the device type. IoT device manufacturers can join the ioXt Certification Program to certify their products. This certification requires verification of the security standards, which can be conducted by third-party companies, authorized test labs, or through self-certification by the manufacturer, who may also offer a reward for identifying security flaws.

A summary of the standards considered in the paragraph above is given in Table 1.

## 3. Preliminaries

The Analytic Hierarchy Process (AHP) [29,30] is a decision-making technique that evaluates multiple alternatives using quantitative and qualitative criteria, even when information is incomplete. Consider *n* alternatives, each associated with an unknown positive value *w_i_ >* 0, and pairwise comparisons A*_ij_* = *ϵ_ij_ ^wi^*, where *ϵ_ij_ >* 0 represents the estimation error. The goal is to estimate the utility vector **w** = [*w*_1_, *w*_2_, …, *w_n_*]*^T^* using a subset of the comparisons, ensuring consistency with A*_ij_*A*_ji_* = 1 and $A_{ji}=A_{ij}^{-1}=\frac{e_{ij}^{-1}\omega_{j}}{\omega_{i}}$.

This problem is modeled with an undirected graph *G* = {*V*, *E*}, where each alternative corresponds to a node *v_i_* ∈ *V*, and a pairwise comparison (*v_i_*, *v_j_*) ∈ *E* defines an edge. The pairwise matrix A has entries A*_ij_*, with A*_ij_* = 0 if (*v_i_*, *v_j_*) ∈/*E*. Among the methods for estimating **w**, the Incomplete Logarithmic Least Squares approach [31,32,33] minimizes the error in logarithmic space. It aims to compute **w***, the logarithmic least squares approximation of **w**, by solving:(1)w*=arg minx ϵR*n⁡12∑i=1n∑jϵNiln⁡Aij−ln⁡xiXj2.

To address this optimization problem, let *y* = ln(*x*), where ln(·) is the component-wise logarithm. This transforms Equation (2) into the following form:(2)w*=arg minx ϵR*n⁡12∑i=1n∑jϵNiln⁡Aij−yi+yj2,
where exp(·) is the component-wise exponential. Defining$$k\left(y\right)=\frac{1}{2}\sum_{i=1}^{n}{\sum_{j\epsilon N_{i}}\left(\ln\left(A_{ij}\right)-y_{i}+y_{j}\right)}^{2},\;$$
and substituting ***y*** = ln(***x***), the problem becomes convex and unconstrained, with the global minimum of the form **w*** = exp(*y**), where *y** satisfies$${\left.\frac{\partial k\left(y\right)}{\partial y_{i}}\right|}_{y=y^{*}}=\sum_{j\epsilon N_{i}}\ln\left(A_{ij}\right)-{y_{i}}^{*}+{y_{j}}^{*}=0,\;\forall i=1,\ldots ,\;n\;,\;$$
indicating the need to find the argument of the function that nullifies its derivative. Moreover, defining the *n* × *n* matrix *P* such that *P_ij_* = ln(A*_ij_*) if A*_ij_ >* 0, and *P_ij_* = 0 otherwise, these conditions can be compactly expressed as:
(3)$$Ly*={P1}_{n},$$
where *L* is the Laplacian matrix of graph *G*. Since *G* is undirected and connected, the Laplacian matrix *L* has rank *n* − 1 [34]. To compute a vector *y* that satisfies this equation, we fix an arbitrary component of *y* and solve a reduced-size system by inverting the resulting nonsingular (*n* − 1) × (*n* − 1) matrix [35]. The vector ***y**** can also be expressed as the arithmetic mean of vectors derived from the spanning trees of the comparison graph, corresponding to the incomplete additive Pairwise Comparison Matrix (PCM) in A [35]. Finally, it is notable that when the graph *G* is connected, the differential equation asymptotically converges to ***y**** (see [36]), offering another approach for its computation.$$\dot{y}\left(t\right)=-Ly\left(t\right)+P1_{n}$$

## 4. Framework of Security for Telemedicine Platform

In this section, we present our new framework of security for telemedicine platforms. The goal was to identify the specific requirements for telemedicine platforms needed to ensure their cybersecurity. To accomplish this, the aspects emphasized by the different standards illustrated in Section 2 were considered to provide a general framework. A schematic of the proposed framework is shown in Figure 1.

The model is realized considering the following steps:
-Analysis of cybersecurity controls provided by relevant regulations and standards, including those specific to medical devices and those related to cybersecurity and data protection;-Collection of security controls deemed necessary for the purpose within a single framework;-Defining the process for verifying the security of a telemedicine platform using the obtained framework as a guide, through evaluation and scoring of individual security controls.

From the analysis of the regulations and guidance presented in the previous section, seventeen controls of the Security Framework for Telemedicine Platforms were identified that can best synthesize and group all the elements present. The different controls were identified based on the cybersecurity elements considered in the different standards. The same were divided into four categories based on the structure of the National Cybersecurity and Data Protection Framework [27]. The subdivision was made as shown below:Identity: analyze the corporate operating context. Resources and investments in line with strategic and risk management objectives are considered.
(a)Risk based approach: Identify, evaluate, and treat the cybersecurity risks associated with the operation of the organization;(b)Regulatory compliance: Software complies with relevant regulations and standards;(c)Asset management: Data, personnel, devices, and systems and facilities are specified and managed consistently with the organization’s objectives and risk strategy;(d)3rd Parties Security: Strict controls are applied on the purchase of components to ensure safety and reliability while minimizing the area of attachment.Protect: The implementation of measures to safeguard business processes and assets. (a)Access control: implementation of robust authentication measures;(b)Data protection: personal data treated with processes defined according to the relevant regulations;(c)Updates and patch management: Software is updated with security patch and bug correction to mitigate known vulnerabilities;(d)Education and awareness: users are made aware of cybersecurity and are trained to fulfill their tasks and roles;(e)Decommissioning: secure decommissioning is ensured, including secure deletion of sensitive data;(f)Network segmentation: the network is divided into separate logical or physical segments in order to limit the possible propagation of a cyber attack.Detect: implement focused action to identify cybersecurity incidents.(a)Monitoring and recording of activities: Registration and monitoring of the users activities;(b)Test and validation: Testing and validation of the software;(c)Application security: The applications conform to the governance;(d)Hardware testing: Evaluation of the hardware device security.Respond and recover: manage the cybersecurity incidents. It intervenes to contain the impact and restore affected processes and services in a timely manner.
(a)Monitoring and recording of security events: Any abnormal activities are detected, and their potential impact is analyzed;(b)Anomalies and security events management: The software can be managed, monitored, and updated over time;(c)Maintainability: Actions are performed to prevent the expansion of a security event, to mitigate its effects, to resolve the incident and to ensure recovery of the affected systems or assets.

To evaluate the relevance of each control, a weight has been assigned to each of them. Specifically, it assigned a weight (column “Control weight (*CW_i_*)”, Table 5) determined by the sum of two factors:

$A_{1}$: Related to the relevance assigned by the standards included illustrated in Section 2. This is achieved by considering how many standards the specific control is enumerated in ($o_{j}^{i}$) and considering the relevance for telemedicine platform of the specific standard ($p_{j}$). This is calculated by multiplying the weight associated with each standard by 1 if the control is present or 0 if it is not present and then dividing the total by the number of standards.

$A_{2}$: Related to the intrinsic relevance of the control itself, given by the product of the weight assigned to the relevant category and the weight of the individual control.

According to the formula, the weight assigned to the *i*-th control belonging to the *h*-th category is as follows:(4)$$Control\;weight\left({CW}_{i}\right)=A_{1}^{i}+A_{2}^{i}=\frac{\sum_{j=1}^{N}o_{j}^{i}*p_{j}}{N}+C_{hi}*s_{i}^{h}$$

$o_{j}^{i}$ is 1 if the *i*-th control is present within the *j*-th standard, 0 otherwise;

$p_{j}$ is the weight assigned to the *j*-th standard;

$C_{hi}$ is the weight assigned to the category *h*-th to which the *i*-th control belongs;

$s_{i}^{h}$ is the weight assigned to the *i*-th control of category *h*-th;

*N* is the number of standards considered, in our case *N* = 10.

To calculate the weights *C_hi_* and $s_{i}^{h}$, the Analytic Hierarchy Process (AHP) methodology developed by Thomas Saaty [30] was used, which allows one to assign a representative value of the degree of preference for each of the given alternatives and use these values to rank and select the alternatives based on a hierarchical structure (see Section 3). Specifically, eight experts were asked to specify the relative importance that various controls have in pairs. The selected experts come from both academic and private backgrounds, presenting experience on the subject matter, as university professors in cybersecurity and security management. Then, for each identified control, a weight was calculated, indicating its importance in relation to the other parameters. Specifically, according to AHP methodology, to evaluate *p_j_* we asked the experts to provide their opinion about the relevance for telemedicine of the different standards illustrated in Section 2. Specifically, they were invited to compare a couple of standards, specifying if standard A is more relevant than standard B using a seven level grade ranging from absolutely more relevant to absolutely less relevant. Notice that due to the large heterogeneity of the standards, the expert must consider only that pair of standards for whom he/she was confident to be able to provide a consistent comparison. The same procedure was performed to estimate *C_hi_* and $s_{i}^{h}$, asking the expert to provide their opinion about the importance of each category and each control, respectively. Also, in these cases, the comparisons have to be performed considering a pair of category/control and limited to those pairs of category/control for which the expert is assumed to be able to provide a consistent pairing.

The scores both for the categories and for the controls are reported in Table 2.

Moreover, in Table 3, we report the values associated with the standards under consideration (*p_j_*) illustrated in Section 2.

In Table 4, we report the occurrence $o_{j}^{i}$.

In order to assess the cybersecurity posture of a telemedicine platform, one has to consider the level of maturity of the adopted controls to estimate the Cyber Risk Posture (CRP) index, i.e.,$$CRP=\sum_{i}{CW}_{i}V_{i}$$
where *V_i_* is the score assigned to the *i*-th control. Specifically, each control can be given a rating ranging from 0 to 5 (“Value” column, Table 5), based on the degree to which it adheres to the relevant requirements, using the CMMI Appraisal Method as the evaluation method [37]. The CMMI Appraisal Method is a structured evaluation process used to assess and improve an organization’s processes based on the Capability Maturity Model Integration (CMMI) framework. It identifies strengths and weaknesses, provides feedback for process enhancement, and determines the organization’s maturity or capability level. There are three types of appraisals: Class A (most rigorous), Class B, and Class C (least rigorous), each serving different purposes such as formal certification or internal assessment. This method helps organizations improve process quality, reduce risks, and increase customer satisfaction.

In particular, it can be inferred that our index allows us to have, in a single value, both the overall risk index (CRP) and to reconstruct a radar plot that allows us to evaluate the different indices for each category. Notice that CRP is useful to assess the overall cyber posture of a telemedicine platform, but it does not provide useful inputs on how to improve the posture. More information can be achieved by using, as illustrated in Figure 2, a radar plot, which allows us to emphasize which controls to improve.

## 5. Case Study

In this section, we analyze a specific case study, considering the ADiLife telemedicine platform [38], developed by ADiTech S.r.l.-Advanced Digital Technologies (Ancona, Italy). Established in 2006, ADiTech serves as a leading entity in creating and integrating innovative solutions for telemedicine, home care, and wellness. ADiLife is a medical software device built on an advanced telemedicine platform that facilitates data sharing and analysis between patients and healthcare providers. The platform integrates various IoT medical devices to monitor critical health parameters such as heart rate, single-lead electrocardiogram (ECG), blood pressure, glucose levels, blood oxygen saturation level (SpO_2_), weight, cholesterol, hemoglobin, and flowmetry, among others. This functionality supports the management of a broad spectrum of conditions, including diabetes, Chronic obstructive pulmonary disease (COPD), cardiovascular diseases, gastrointestinal disorders, oncology, Amyotrophic lateral sclerosis (ALS), and spinal muscular atrophy (SMA).

Following the development of the Security Framework for telemedicine platforms, a comprehensive analysis was conducted to evaluate the cybersecurity aspects of the ADiLife and ADiTech S.r.l. platforms. This assessment aimed to identify potential vulnerabilities and areas for improvement, ensuring the security of the medical devices and compliance with the controls defined by the Framework. The evaluation is presented through the CRP parameter established in the Framework. The values for CW are determined based on the relevant standards and the results of the analysis obtained using the AHP methodology. Further details on the assignment of the *V_i_* values associated with each weight will be discussed below, focusing on their relation to the implementation of platform security measures.

Based on the risk-based approach, the risk management policy for the ADiLife medical software device adheres to the Risk Management Plan, incorporating risk management activities into the design, development, and deployment processes. This approach aligns with ISO 14971 standards, which dictate risk management for medical devices. The risk management process encompasses several stages: risk analysis (identification of intended use and associated hazards, risk estimation), risk evaluation, risk control, overall residual risk evaluation, risk management review, and production and post-production monitoring of risk control efficacy.

Specific responsibilities within the risk management process are allocated to designated roles: Risk Management Process Manager, Risk Management Phase Manager, Risk Mitigation Measures Verifier, and Context Expert.

Three-level scales are used to define probability and severity, leading to a risk estimation matrix that guides the acceptability of different risk levels.

Risk control measures are implemented for all non-acceptable risks to mitigate their probability and severity. Post-control, the overall residual risk is re-evaluated, considering the effectiveness of the control measures. Prior to the medical device’s commercial release, a thorough risk management review ensures proper implementation, residual risk acceptability, and appropriate post-production monitoring methods. Any new risks identified during post-production trigger a comprehensive risk reassessment. In total, the value for the “Risk based approach” is 5.

Considering “Access Control”, access to the platform is secured through username and password authentication, as it is crucial to avoid any unwanted access to certain platform resources, which could compromise the security of sensitive patient data. In addition, an access control system is in place that uses REST API call controls to prevent unauthorized access and detect potential attacks. Anomalous patterns trigger progressive user blocking measures. Therefore, the score for “Access control” is 5.

It is continued by considering “Data Protection”. To reduce the risks of unauthorized access or misuse of personal data, the principle of data pseudonymization is applied, which involves storing data in a form that prevents the identification of the individual without the use of additional information. Data pseudonymization and encryption using the SHA-256 algorithm safeguard patient data. As for data traffic over the public network, this is protected via https/SSL protocol; therefore, data transmission from the App to the portal and vice versa is also conducted securely. For this reason, the score is 5.

The next field to analyze is “Software Updates and Patch Management”. Regular maintenance and updates ensure software security, with at least six updates per year for the mobile application. Therefore, the value for this field is 5.

Moreover, we analyze “Activity Monitoring and Logging”. It is implemented through comprehensive monitoring and recording of user activities. For this reason, all REST API calls are logged and analyzed for consistency. The score is 5.

Another important aspect, “Training and Awareness”, is considered. Ongoing cyber-security training is provided to healthcare operators, while patient training is managed by healthcare facilities. As such, this aspect was evaluated 5.

The software complies with EU MDR, ISO 13485, and GDPR regulations, ensuring adherence to relevant safety and data protection standards. Therefore, “Regulatory Compliance” has a score of 5.

Detection of any abnormal activity is conducted at the same time as recording and analyzing the REST calls that the system receives. Potentially risky activities trigger access endpoint closures and necessary verifications, with standard GDPR procedures applied in case of data breaches. There is also a person in charge of the platform’s IT security and the definition of security procedures and verification of their implementation. Security Event Management was evaluated 5.

Analyzing “Secure Decommissioning”, data deletion is managed exclusively by ADiTech upon explicit request from healthcare entities, following a B2B2C distribution model. The evaluation is 5.

Internal security vulnerability analyses are conducted with each update, though external third-party security testing is not performed. So “Test and Validation” has a score of 1.

Considering “Asset Management”, data acquisition is minimal and aligned with organizational risk strategy, compliant with MDR and GDPR. The evaluated score is 5.

Medical devices and their integration APIs are certified. In addition, third-party security is verified also by healthcare entities. So, the “Third-Party Security” is evaluated 5.

Considering “Maintainability”, the software undergoes regular maintenance and updates, ensuring long-term management and monitoring. Therefore, the score for this parameter is 5.

For the “Application security”, the same applies as for “Test and validation”, and consequently the score is 1.

Analyzing “Network Segmentation”, the platform architecture is composed of a front-end part and a back-end part. The front-end includes an app and a web portal published on cloud infrastructure, accessible to caregivers and patients. The back-end part resides in a restricted area of the cloud infrastructure, responsible for access administration, data storage, management, and fruition, accessible only through private connections. “Network segmentation” was evaluated with the highest score (5).

Lastly, “Hardware Testing” was evaluated with a score of 0, because specific hardware tests are not conducted.

All assigned scores and weighted values obtained for each control are shown in Table 5, where the control weight represents the value obtained with our approach and the weighted value is the value obtained for the combination of our index and the value associated with each security control. On the basis of such data, we can estimate the CRP for the ADiLife platform to be equal to 10.37. This represents a very high level, hence the platform can be considered to have a high cyber posture. For more detail, we summarized the results in a radar plot (see Figure 2) that shows the weighted values.

## 6. Discussion

Cybersecurity in the healthcare sector, as previously highlighted, is a critically important issue, particularly with the growing prevalence of tools and systems that connect patients and their sensitive data with various services. Our work introduces a new approach for managing and evaluating the cybersecurity aspects of a telemedicine platform. As discussed in the literature, several approaches address cybersecurity in healthcare.

Unlike the solution presented in [14], our work is not limited to analyzing aspects related to the protection of sensitive patient data. While [19] focuses on applying blockchain to address trust and security issues—a relevant approach for mitigating risks—our framework provides a broader analysis of various risks and solutions to safeguard against cyberattacks. Additionally, our solution is designed specifically for medical informatics applications and aims to meet regulatory and compliance requirements, a critical aspect of telemedicine platforms but not the sole focus of our analysis. The framework we propose could incorporate blockchain as a mitigation strategy following the risk evaluations performed using our approach.

The framework proposed in [15], like ours, employs security assessments based on multicriteria decision-making methods such as AHP (Analytic Hierarchy Process) and TOPSIS (Technique for Order of Preference by Similarity to Ideal Solution). This method enables a systematic and quantitative evaluation of vulnerabilities and comparisons of different security strategies. However, our framework goes further by including a detailed analysis of the telemedicine platform’s compliance with relevant standards and regulations. Compliance weighting in our framework is derived not only from MCDM analysis but also from an objective assessment of adherence to these regulations. While [15] is particularly suited for connected health devices, such as sensors and wearables, this domain represents a potential area where our framework could also be tested and validated.

The cybersecurity framework described in [13] is designed for cyber-physical systems in medical environments, encompassing interconnected medical devices like infusion pumps and medical imaging equipment. While these systems differ from telemedicine platforms, our framework could potentially be adapted and tested in this context as well.

For telemedicine applications, the framework presented in [10] introduces a risk management method to identify, analyze, and mitigate security risks associated with these systems. This approach employs an “attack tree” method to provide a structured response to potential attacks. In contrast, our framework emphasizes proactive analysis to optimize and improve the system before threats materialize, offering a preventative strategy in addition to mitigation.

In conclusion, our proposed method provides a quantitative system for evaluating the quality and security of telemedicine platforms. Future work may involve expanding case studies and exploring the feasibility of applying the same method to other types of remote patient interaction tools.

## 7. Conclusions

The analysis revealed that cybersecurity in telemedicine platforms is a growing concern, as the healthcare sector becomes increasingly vulnerable to cyber threats. The research underscored the importance of proactive approaches to managing cybersecurity in telemedicine platforms, given their critical role in delivering healthcare services. Investing in cultivating a security-oriented corporate culture and adopting appropriate resources and technologies are essential steps to ensure the protection of sensitive data and the continuity of operations in telemedicine, as well as safeguarding patient health.

Currently, with the rapid expansion of these tools and the significant impact that potential cyberattacks can have, awareness of the problem has grown significantly. This is evidenced by the allocation of funds in the National Recovery and Resilience Plan (NRP) for telemedicine and the introduction and adoption of specific cybersecurity standards tailored to the healthcare sector.

In this context, the Security Framework for Telemedicine Platforms developed during the study serves as a valuable tool for assessing the security of telemedicine platforms and identifying vulnerabilities and areas for improvement. This framework enables companies to strengthen their cybersecurity strategies effectively.

## Figures and Tables

**Figure 1 healthcare-13-00184-f001:**
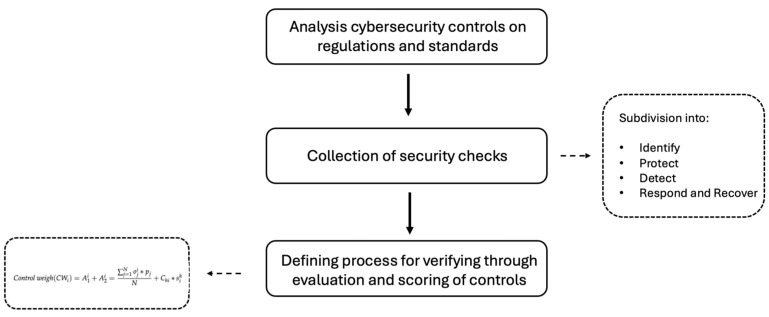
Outline of the proposed framework.

**Figure 2 healthcare-13-00184-f002:**
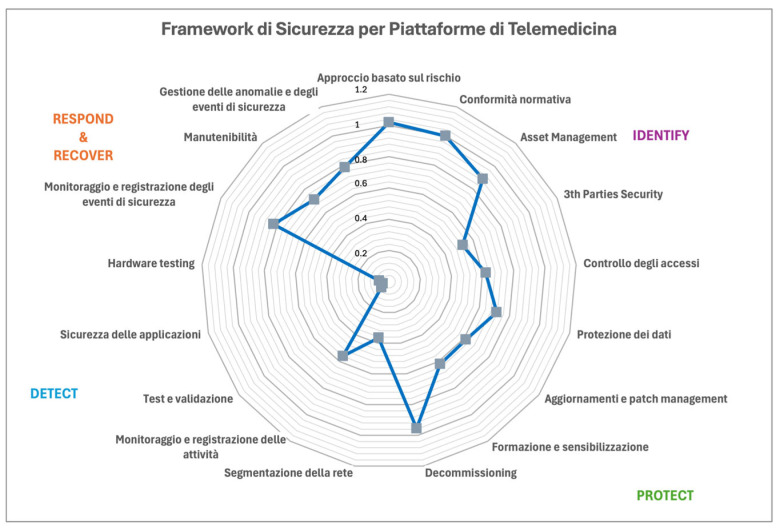
Radar plot summarizing the different components.

**Table 1 healthcare-13-00184-t001:** Summarization of the measure presented in the state of the art.

Regulation	Explanation
MDR 745:2017 [20]	This regulation aims to establish how it proceeds to uphold high safety standards.
EN ISO 13485 [21]	Practical foundation for addressing the MDR’s requirements.
ISO 14971 [22]	It specifies the main concept for risk management
IEC 62304 [23]	Development and lifecycle management of medical devices software
IEC 81001-5-1 [24]	Guideline for developing and maintaining secure medical devices
GDPR [25]	Rules for processing personal data
ISO 27100 [26]	Guideline for information Security management system
The Italian national Framework [27]	It aims to manage security and regulatory compliance
ioXt Alliance [28]	Consortium to develop and promote guidelines, standards, and certification

**Table 2 healthcare-13-00184-t002:** Categories and controls weights.

Category	Category Weight *C_hi_*	Controls	$\mathrm{Control}\;\mathrm{Weight}\;s_{i}^{h}$
Identify	0.40	Risk based approach	0.34
		Regulatory compliance	0.34
		Asset management	0.26
		3rd Parties Security	0.06
Protect	0.30	Access control	0.14
		Data Protection	0.14
		Updates and Patch management	0.14
		Education and awareness	0.10
		Decommissioning	0.42
		Network segmentation	0.06
Detect	0.13	Monitoring and recording of activities	0.23
		Test and Validation	0.13
		Application security	0.23
		Hardware testing	0.40
Respond and Recover	0.17	Monitoring and recording of security events	0.33
		Anomalies and security events management	0.33
		Maintainability	0.33

**Table 3 healthcare-13-00184-t003:** Standards weights.

Standard	*p_j_*
MDR 745:2017	0.11
EN ISO 13485	0.07
EN ISO 14971	0.03
EN IEC 62304	0.05
GDPR	0.10
IEC 81001-5-1	0.09
IEC 27100	0.09
ioXt Alliance	0.07
Italian National Framework for Cybersecurity and Data Protection	0.14

**Table 4 healthcare-13-00184-t004:** Occurrence matrix for control into standard.

Security Controls	MDR745:217	EN ISO 13485	EN ISO 14971	EN ISO 62304	GDPR	IEC 81001-5-1	IEC 27100	IoXt Alliance	National Framework	$o_{j}^{i}$
Risk based approach	1	1	1	1	1	1	1	0	1	8
Regulatory compliance	0	0	0	0	0	0	1	0	0	1
Asset management	0	0	0	0	0	0	1	0	1	2
3rd Parties Security	0	1	0	0	0	0	1	0	1	3
Access control	0	0	0	0	0	1	1	0	1	3
Data Protection	0	0	0	0	1	1	1	1	1	5
Updates and Patch management	0	0	0	0	0	1	1	1	0	3
Education and awareness	1	1	0	0	0	1	1	0	1	5
Decommissioning	0	0	0	0	0	0	0	0	0	0
Network segmentation	0	0	0	0	0	0	0	0	0	0
Monitoring and recording of activities	0	0	0	0	1	1	1	0	0	3
Test and Validation	0	0	0	1	0	0	1	1	1	4
Application security	0	0	0	0	0	0	0	0	0	0
Hardware testing	0	0	0	0	0	0	0	0	0	0
Monitoring and recording of security events	1	1	0	1	1	1	1	0	1	7
Maintainability	1	0	0	0	0	1	1	1	0	4
Anomalies and security events management	1	1	0	1	1	0	1	0	1	6

**Table 5 healthcare-13-00184-t005:** Structure of the security framework for the telemedicine platform with the results of the ADiLife Assessment.

Category	Control	*CRPi*	*CWi*	Value
Identify	Risk based approach	1.02	0.20	5
	Regulatory compliance	1	0.2	5
	Asset management	0.89	0.18	5
	3rd Parties Security	0.52	0.10	5
Protect	Access control	0.62	0.12	5
	Data Protection	0.71	0.14	5
	Updates and Patch management	0.62	0.12	5
	Education and awareness	0.62	0.12	5
	Decommissioning	0.96	0.19	5
	Network segmentation	0.37	0.07	5
Detect	Monitoring and recording of activities	0.56	0.11	5
	Test and Validation	0.06	0.06	1
	Application security	0.04	0.04	1
	Hardware testing	0.06	0.06	1
Respond and Recover	Monitoring and recording of security events	0.82	0.16	5
	Anomalies and security events management	0.71	0.14	5
	Maintainability	0.78	0.16	5

## Data Availability

Data are contained within the article.

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
