# Peer review of "Cybersecuity Analysis of a Telemedicine Platform"

_healthcare, 2025, doi:10.3390/healthcare13020184_

Round 1

Reviewer 1 Report

Comments and Suggestions for Authors

1. The first paragraph of the introduction section should include a relevant and authoritative reference to substantiate your statements and provide a stronger foundation for your research context.

2. Consider presenting your research methodology or procedural steps in a visual format, such as flowcharts, diagrams, or tables. These elements will enhance clarity, make complex processes more understandable, and improve the overall readability of your paper.

3. Add a dedicated discussion section that compares your research results with findings from previous studies, particularly focusing on a security framework for telemedicine platforms. This will contextualize your findings, emphasize their significance, and identify potential areas for future research advancements.

Author Response

We would like to thank the Reviewer for their thorough revision and valuable observations that helped usimprove the earlier version of the paper. It is always extremely helpful to have an external opinion on sectionsthat need clarification. We fundamentally agree with all the comments made by the Reviewer, and we have incorporated corresponding revisions into the revised version of the manuscript. The suggestions provided have been precious in the re-writing of the paper, as we hope it emerges from the examination of the present document. Below, it is possible to find our point-by-point response to reviewers’ comments. It is our feeling that along the line of reviewing the paper, we managed to improve it in both presentation and readability. The changes are implemented in the revised draft of the article in red.

Comment 1: The first paragraph of the introduction section should include a relevant and authoritative reference to substantiate your statements and provide a stronger foundation for your research context.

Answer 1: Thanks to the reviewer for the comment, some references have been added in order to substantiate the claims within the paragraph.

Comment 2: Consider presenting your research methodology or procedural steps in a visual format, such as flowcharts,diagrams, or tables. These elements will enhance clarity, make complex processes more understandable, and improve the overall readability of your paper.

Answer 2: We thank the reviewer for the comment, a visual diagram of the work process summarizing the steps ofthe presented framework has been added

Comment 3: Add a dedicated discussion section that compares your research results with findings from previous studies, particularly focusing on a security framework for telemedicine platforms. This will contextualize your find- ings, emphasize their significance, and identify potential areas for future research advancements.

Answer 3: We thank the reviewer for the comment, a discussion section has been added

Reviewer 2 Report

Comments and Suggestions for Authors

Summary

The rapid expansion of telemedicine during the COVID-19 pandemic has introduced critical cybersecurity vulnerabilities in securing sensitive medical data and ensuring the secure operation of telemedicine platforms. These vulnerabilities, if not properly addressed, can compromise patient safety and the integrity of healthcare systems. The proposed framework presents a structured approach to implement cybersecurity measures in telemedicine platforms, drawing from the most relevant elements recognized in the most adopted standards for medical devices. The framework allows for assessing the cyber threats of a telemedicine platform and provides concrete elements on how to improve it, as demonstrated through a case study.

However, some aspects need to be addressed for value addition for the target readers and to strengthen the current version of manuscript.

Comments for improvements:

1.      The abstract should clearly present the outcomes/benefits of the proposed system.

2.      The keywords should be updated according to the theme of manuscript

3. The introduction is too lengthy and wordy. It should be rearranged for comprehending.

4.    The state of the art must be accumulated/ converted into a tabular form to enhance readability and understanding.

5.   A graphical representation of section 3. Framework of security for telemedicine platform is required.

6.   The measurable effects of control weights should be justified in the results and analysis section to know and understand its impact and consequences.

7.     The most important point is the analysis of the case study is missing in terms of the parameters selected/chosen.

8.      The conclusion should be consistent with the abstract.

Author Response

We would like to thank the Reviewer for their thorough revision and valuable observations that helped usimprove the earlier version of the paper. It is always extremely helpful to have an external opinion on sections thatneed clarification. We fundamentally agree with all the comments made by the Reviewer, and we have incorporated corresponding revisions into the revised version of the manuscript. The suggestions provided have been precious in the re-writing of the paper, as we hope it emerges from the examination of the present document. Below, it is possible to find our point-by-point response to reviewers’ comments. It is our feeling that along the line of reviewing the paper, we managed to improve it in both presentation and readability. The changes are implemented in the revised draft of the article in red.

Comment 1 The abstract should clearly present the outcomes/benefits of the proposed system.

Answer 1 We thank the reviewer for the comment, we implement the abstract to present better the benefits of the platform.

Comment 2 The keywords should be updated according to the theme of manuscript Answer 2.2 We thank the reviewer for this comment, we added further keywords and we modify some of the previously.

Comment 2.3 The introduction is too lengthy and wordy. It should be rearranged for comprehending

Answer 3 We thank the reviewer for the comment, the introduction was modified in order to make it less wordy  and more organized for understanding.

Comment 4 The state of the art must be accumulated/ converted into a tabular form to enhance readability and understanding.

Answer 4 We thank the reviewer, we add a table that summerized the state of the art.

Comment 5 A graphical representation of section 3. Framework of security for telemedicine platform is required.

Answer 5 We thank the reviewer, a diagram has been added summarizing the steps that define the framework

Comment 6 The measurable effects of control weights should be justified in the results and analysis section to know and understand its impact and consequences.

Answer 6 As better explained in the text, the effect of control has been estimated by a set of experts  and the relative weight has been calculated solving an AHP problem

Comment 7 The most important point is the analysis of the case study is missing in terms of the parameters selected/chosen.

Answer 7 We would like to thank the reviewer for the comment, an attempt has been made to better explain what elements are analyzed within the case study.

Comment 8 The conclusion should be consistent with the abstract.

Aswer 8 We thank the reviewer for the comment the conclusions and abstract have been edited to make them consistent

Reviewer 3 Report

Comments and Suggestions for Authors

This manuscript is an explanatory study that presents the necessary framework and elements for implementing cybersecurity measures in building telemedicine platforms. It also provides a case study that assesses the security level of an actual telemedicine platform using the proposed framework. Through this, the study aims to enhance the quality of medical services that handle patient information remotely and to support the effective establishment of policies for telemedicine platforms.

Please refer and clarify below comments:

0. Introduction

1) In the first paragraph, it is stated that the demand for telemedicine has increased due to COVID-19 and that it has accelerated advancements in new technologies within the healthcare sector. However, references to support these statements are necessary.

2) In the second paragraph, it is mentioned that the fund for telemedicine has increased by 500 million under Italy's PNRR policy. Please specify the exact currency of this budget.

1. State of the art

1) In section 1.1. Medical regulations, please provide a reference for the explanation of IEC 81001-5-1, ISO/IEC 27001, ioXt Alliance (Internet of Secure Things Alliance).

2. Preliminaries

1) The entire content presented in the Preliminaries section is identical to that of the paper listed below. Please summarize the Preliminaries content concisely and cite the corresponding paper as a reference.

M. Nobili et al., "DRIVERS: A platform for dynamic risk assessment of emergent cyber threats for industrial control systems," 2023 31st Mediterranean Conference on Control and Automation (MED), Limassol, Cyprus, 2023, pp. 395-400, doi: 10.1109/MED59994.2023.10185686.

3. Framework of security for telemedicine platform 3

1) While the framework claims to be new, it lacks explicit justification or comparison with existing frameworks in section 1. State of art to demonstrate its novelty and added value.

2) There is no explanation of why the specific seventeen controls were chosen or how they address the unique challenges of telemedicine platforms.

3) Please provide a reference for the structure of the National Cybersecurity and Data Protection Framework which was used to divide four categories with seventeen controls of the Security Framework for Telemedicine Platforms.

4) In lines 396-399, it is mentioned that experts were consulted to evaluate the importance of each control. Please provide detailed information about the careers and qualifications of the participating experts, such as "professor in cybersecurity," etc.

5) In lines 415, please provide the number of Table which indicates the occurrence.

6) In lines 415, please provide the number of Table which indicates the occurrence.

7) Please format the titles of Tables 1, 2 and 3 according to the journal manuscript style and position them above the tables.

4. Case Study

1) Please provide a reference for the ADiLife telemedicine platform used in the Case Study.

2) In lines 441-445, when using abbreviations related to health parameters such as ECG, COPD, ALS and SMA, please provide the full name alongside the abbreviation.

3) In Figure 1 and Table 4, the security level of the ADiLife telemedicine platform is presented based on the new evaluation framework. To assess whether the results of the new framework are appropriate, please provide a comparison with the results obtained by evaluating the ADiLife platform using other state-of-the-art frameworks.

Others

1) This study descriptively presents a new framework for evaluating the security level of telemedicine platforms. However, before presenting the Conclusion, it is necessary to add a Discussion section that compares the results of this study with previous studies on other frameworks. Additionally, the limitations of this study should be detailed in the Discussion section.

Author Response

We would like to thank the Reviewers for their thorough revision and valuable observations that helped us improve theearlier version of the paper. It is always extremely helpful to have an external opinion on sections that need clarification. We fundamentally agree with all the comments made by the Reviewers, and we have incorporated corresponding revisions into the revised version of the manuscript. The suggestions provided have been precious in the re-writing of the paper, as we hope it emerges from the examination of the present document. Below, it is possible to find our point-by-point response to reviewers’ comments. It is our feeling that along the line of reviewing the paper, we managed to improve it in both presentation and readability. The changes are implemented in the revised draft of the article in red.  

Comment 1 In the first paragraph, it is stated that the demand for telemedicine has increased due to COVID-19 and that it has accelerated advancements in new technologies within the healthcare sector. However, references to support these statements are necessary.

Answer 1 We thank the reviewer references have been added to support the statement.

Comment 2 In the second paragraph, it is mentioned that the fund for telemedicine has increased by 500 million under Italy’s PNRR policy. Please specify the exact currency of this budget.

Answer 2 Thanks to the reviewer for the comment, the budget for the measure has been added.

Comment 3 In section 1.1. Medical regulations, please provide a reference for the explanation of IEC 81001-5-1, ISO/IEC 27001, ioXt Alliance (Internet of Secure Things Alliance).

Answer 3 We thank the reviewer for the comment, we add the reference for the standard regulation reporting above.

Comment 4 The entire content presented in the Preliminaries section is identical to that of the paper listed below.Please summarize the Preliminaries content concisely and cite the corresponding paper as a reference.

  1. Nobili et al., ”DRIVERS: A platform for dynamic risk assessment of emergent cyber threats for industrial control systems,” 2023 31st Mediter- ranean Conference on Control and Automation (MED), Limassol,Cyprus, 2023, pp. 395-400, doi:10.1109/MED59994.2023.10185686.

Answer 4 We thank the reviewer, we modify the preliminaries paragraph. 

Comment 5 While the framework claims to be new, it lacks explicit justification or comparison with existing frameworks in section 1. State of art to demon- strate its novelty and added value.

Answer 5 We appreciate the reviewer for the comment we added a comparison with other frameworks within the state of the art and within the discussion section.

Comment 6 There is no explanation of why the specific seventeen controls were chosen or how they address the unique challenges of telemedicine platforms.

Answer 6 We would like to thank the reviewer for the comment a brief explanation of the choice of different criteria has been added.

Comment 7 Please provide a reference for the structure of the National Cybersecurity and Data Protection Framework which was used to divide four categories with seventeen controls of the Security Framework for Telemedicine Platforms.

Answer 7 We thank the reviewer, we add a reference.

Comment 8 In lines 396-399, it is mentioned that experts were consulted to evaluate the importance of each control.Please provide detailed information about the careers and qualifications of the participating experts, such as ”professor in cybersecurity,” etc.

Answer 8 We thank the reviewer for the comment, we have added more explanation of the background of the experts.

Comment 9 In lines 415, please provide the number of Table which indicates the oc- currence.

Answer 9 We thank the reviewer for the comment, the table number referring to the occurrence was inserted.

Comment 10 In lines 415, please provide the number of Table which indicates the oc- currence.

Answer 10 We thank the reviewer for the comment, the table number referring to the occurrence was inserted.

Comment 11 Please format the titles of Tables 1, 2 and 3 according to the journal manuscript style and position them above the tables.

Answer 11 Thanks to the reviewer for the comment, the captions of Tables 1,2,3 have been changed and moved above the table

Comment 12 Please provide a reference for the AdiLife telemedicine platform used in the Case Study.

Answer 12 We thank the reviewer for the comment, we add a reference for the plat- form.

Comment 13 In lines 441-445, when using abbreviations related to health parameters such as ECG, COPD, ALS and SMA, please provide the full name along- side the abbreviation.

Answer 13 We thank the reviewer, we add the full name associate with the abbrevi- ations.

Comment 14 In Figure 1 and Table 4, the security level of the ADiLife telemedicine platform is presented based on the new evaluation framework. To assess whether the results of the new framework are appropriate, pleaseprovide a comparison with the results obtained by evaluating the ADiLife platform using other state-of-the-art frameworks.

Answer 14 As illustrated in the text actually there is no single framework able to collect the different aspect related to cybersecurity of tele-medicine platform. Hence comparison with other frameworks might provide ambiguous result.

Comment 15 This study descriptively presents a new framework for evaluating the security level of telemedicine platforms. However, before presenting the Conclusion, it is necessary to add a Discussion section that compares the results of this study with previous studies on other frameworks. Addi- tionally, the limitations of this study should be detailed in the Discussion section.

Answer 15 We thank the reviewer for the comment, a discussion section has been added